# A Lightweight Diabetic Retinopathy Detection Model Using a Deep-Learning Technique

**DOI:** 10.3390/diagnostics13193120

**Published:** 2023-10-03

**Authors:** Abdul Rahaman Wahab Sait

**Affiliations:** Department of Documents and Archive, Center of Documents and Administrative Communication, King Faisal University, P.O. Box 400, Hofuf 31982, Al-Ahsa, Saudi Arabia; asait@kfu.edu.sa

**Keywords:** diabetic retinopathy, machine learning, MobileNet V3, Yolo V7, deep learning, artificial intelligence

## Abstract

Diabetic retinopathy (DR) is a severe complication of diabetes. It affects a large portion of the population of the Kingdom of Saudi Arabia. Existing systems assist clinicians in treating DR patients. However, these systems entail significantly high computational costs. In addition, dataset imbalances may lead existing DR detection systems to produce false positive outcomes. Therefore, the author intended to develop a lightweight deep-learning (DL)-based DR-severity grading system that could be used with limited computational resources. The proposed model followed an image pre-processing approach to overcome the noise and artifacts found in fundus images. A feature extraction process using the You Only Look Once (Yolo) V7 technique was suggested. It was used to provide feature sets. The author employed a tailored quantum marine predator algorithm (QMPA) for selecting appropriate features. A hyperparameter-optimized MobileNet V3 model was utilized for predicting severity levels using images. The author generalized the proposed model using the APTOS and EyePacs datasets. The APTOS dataset contained 5590 fundus images, whereas the EyePacs dataset included 35,100 images. The outcome of the comparative analysis revealed that the proposed model achieved an accuracy of 98.0 and 98.4 and an F1 Score of 93.7 and 93.1 in the APTOS and EyePacs datasets, respectively. In terms of computational complexity, the proposed DR model required fewer parameters, fewer floating-point operations (FLOPs), a lower learning rate, and less training time to learn the key patterns of the fundus images. The lightweight nature of the proposed model can allow healthcare centers to serve patients in remote locations. The proposed model can be implemented as a mobile application to support clinicians in treating DR patients. In the future, the author will focus on improving the proposed model’s efficiency to detect DR from low-quality fundus images.

## 1. Introduction

DR is a retinal complication of diabetes [1]. It impairs or completely degrades an individual’s vision. Uncontrolled diabetes over an extended length of time increases the risk of visual impairment due to diabetic maculopathy [2,3,4]. Retinal capillaries are susceptible to damage from high blood sugar. In a longer period, this deterioration leaves blood vessels more vulnerable to further damage or even rupture [5]. The risk of DR depends on diabetes duration, blood sugar management, genetic susceptibility, hypertension, and lipid abnormalities [6,7,8]. DR is more likely to develop in type 1 and 2 diabetics with poor blood sugar management [9]. DR is the primary factor of irreversible blindness in individuals across the world [10]. In addition, DR contributes to serious disorders like proliferative DR, the most prevalent microvascular implication [11]. Early diagnosis is one of the crucial factors for reducing the severity of DR.

The field of ophthalmology relies heavily on analyzing blood vessel structures in retinal fundus images. Permanent vision loss can occur due to age-related macular degeneration and diabetic macular edema [11]. Optical Coherence Tomography (OCT) is a crucial tool for ophthalmologists in diagnosing DR [12]. Ophthalmologists must devote a considerable amount of time to detecting abnormalities. This is crucial to prevent or mitigate DR-related visual impairment. Adopting minimally invasive methods and robotic-assisted surgery has become increasingly prevalent in ophthalmology, especially for treatments such as cataract surgery and glaucoma treatment [11]. These advancements have demonstrated the ability to reduce patient discomfort and expedite healing. To increase the efficacy of ocular drugs, new drug delivery devices such as sustained-release implants and punctal plugs have been developed [12]. Utilizing teleophthalmology facilitates both the remote evaluation of retinal images and consultations with patients [12]. It has demonstrated enhanced accessibility to DR screening. Emerging imaging techniques like hyperspectral and multispectral imaging have demonstrated potential in the realm of early disease identification and tissue characterization [12]. Medical diagnosis and therapy have greatly benefited from developments in 3D and 4D medical imaging, which have enhanced the visualization of anatomical structures [12]. The implementation of portable and handheld imaging technologies has experienced a surge in popularity, as it allows healthcare practitioners to conduct imaging activities to provide effective treatments.

In the Kingdom of Saudi Arabia (KSA), it is estimated that 13.4% of individuals are affected by diabetes mellitus, making it an extremely serious medical condition [11,12]. An automated and affordable screening system is required to serve DR patients across Saudi Arabia [12]. Medical and surgical operations for these individuals are more costly, and their unfavorable prognoses impose a financial strain on them and the healthcare system. The Saudi National Diabetes Center was recently founded to address the prevalence and severity of diabetes [12]. The center has spearheaded a strategy plan to significantly enhance diabetes treatment in the Saudi population over the upcoming years. There is a demand for an automated detection model to identify DR in the earlier stages.

In contrast to more traditional procedures, such as the dilatation of the eye pupil, automated retinal image processing has greatly facilitated the diagnosis of retinal diseases [13]. In recent years, artificial intelligence (AI) and machine learning (ML) algorithms have made significant advancements in the automated identification and assessment of DR using retinal images. The primary objective of these systems is to optimize the early detection of medical conditions and increase the overall care and treatment of patients [14]. AI systems can examine retinal images and scans to identify the earliest stages of DR. These algorithms can detect and categorize DR severities [14]. The computerized screening procedure aids in making a timely diagnosis, which is essential for effective therapy. AI applications can help prioritize patients according to the severity of their conditions [15]. It can be used to evaluate large datasets of retinal images and patient information to improve DR detection strategies.

Fundus images, including OCT scans and ultrasonography, are widely applied to DR detection. These images cover blood vessels, the macula, and the retina’s interior part. The fundus camera provides high-quality retinal images. These images are used in deep-learning (DL) models for detecting abnormalities [16].

A convolutional neural network (CNN) is a subset of artificial neural network architecture. It is primarily used in processing videos and images [16]. In recent years, CNNs have played a pivotal role in advancing computer vision by assisting in resolving various visual recognition challenges [17]. CNNs are built from numerous distinct convolutional layers, each of which must learn and identify certain image characteristics or patterns. The computation of feature maps is the goal of convolutional operations, which entail shifting extremely small filters across the input image [18]. It is possible to fine-tune pre-trained CNN models for use in multiple applications. These models have been exposed to extensive data and have gained an enormous amount of knowledge in various feature domains. Transfer learning (TL) approaches present an exceptional outcome using smaller datasets [19]. To enhance feature extraction and prediction for DR detection and to overcome the limitation of unbalanced and noisy fundus image data, existing studies have employed many data-augmentation approaches, sampling techniques, cost-sensitive algorithms, and hybrid and ensemble architectures [20].

The large datasets, including MESSIDOR, EyePacs, and APTOS, provide the fundus images [21]. Ophthalmologists were involved in gathering the ground truth images. The researchers employ these datasets to generalize their DR-detection models [21]. The CNN-based DR-detection models are widely used to detect and grade DR severity. These models demand a higher number of computational resources for producing an outcome. There is a lack of lightweight CNN models for detecting DR severity. This motivated the author to develop a lightweight model for grading the DR-severity level using the fundus images. In addition, an effective mobile-based DR image classifier is required to provide services to the individuals in the remote locations of the KSA.

The contributions of this study are as follows:i.A feature-extraction technique to improve the accuracy of the DR-detection model.ii.A DR-severity grading model that demands fewer parameters, FLOPs, and convolutional layers.iii.An evaluation of the proposed model using the benchmark datasets and evaluation metrics.

This proposed study is structured as follows: Section 2 presents the existing DR-severity literature. Section 3 outlines the proposed methodology for classifying the DR-severity levels. The findings are presented in Section 4. Section 5 discusses this study’s contribution to the DR-detection literature. Finally, Section 6 concludes this study.

## 2. Literature Review

Medical professionals can benefit greatly from deep-learning-based systems that automate the interpretation of retinal pictures and provide objective and consistent assessments of the severity of DR [21]. DL-based screening methods can test a large diabetic population for DR. Deep-learning algorithms can monitor the course of diseases over time by evaluating successive retinal images [21]. As a result, physicians may fine-tune their approaches to treating patients. Implementing these technologies plays an essential role in augmenting the efficacy and proficiency of DR screening and therapy, ultimately yielding advantages for both patients and healthcare practitioners. Nagpal et al. (2022) [22] discussed the recent developments in the DR-detection models. The noise and low contrast levels of the images may reduce the DR-image-classification performance. The morphological changes in the retinal images are the key factors in detecting DR [23]. In addition, DR-detection models identify lesions to compute severity levels.

Orlanda et al. (2017) [23] proposed a DL-based lesion-detection model using ensemble values. Al-hazaimeh et al. (2022) [24] developed a multi-class classification model for detecting DR severity. They followed blood-vessel-based segmentation and optic-disc-based detection techniques for pre-processing the images. In addition, they applied feature extraction and selection techniques to improve the classifier accuracy. Suganyadevi et al. (2022) [25] proposed a DR-detection model for detecting the severity of the fundus images. They employed the CNN models for processing the images. The multi-class classifier achieved an optimal outcome. Similarly, Nahiduzzaman et al. (2023) [26] developed a DR-identification model using a parallel convolutional neural network. They used an extreme learning machine to extract the key patterns. They adjusted the CNN model’s parameters using hyperparameter optimization. They used a smaller number of parameters for classifying the fundus images.

Abbood et al. (2022) [27] developed a hybrid retinal image enhancement algorithm using the DL technique. They applied a retinal-cropping technique to extract the features. Gaussian blur and circle cropping were used to enhance the image quality. They employed a ResNet 50 model to classify the fundus images. Canayaz (2022) [28] proposed a classification technique to detect DR severity. Binary Bat algorithm, Equilibrium optimizer, Gray Wolf optimizer, and Gravity search algorithm were used for feature extraction. They used a Support Vector Machine and Random Forest for classifying DR-severity levels. Modi and Kumar (2022) [29] developed a DR-severity detection using a Bat-based feature selection algorithm. They employed a deep forest technique for image classification. The K-mean-based segmentation algorithm was used to identify the lesion region. The feature extraction was performed using a multi-grained scanning method. Dayanna and Emmanuel (2022) [30] proposed a grading system for identifying the severity levels of the fundus images. The coherence-enhancing energy-based regularized level set evolution was used for blood-vessel segmentation. An attention-based fusion network was employed to detect the candidate lesion region. They applied a deep CNN model to classify the fundus images.

Furthermore, Savelli et al. (2020) [31] employed the multi-context ensemble-based CNN for detecting lesions in the fundus images. Chetoui et al. (2020) [32] employed EfficientNet to identify the abnormalities. Karki et al. (2021) [33] proposed an integrated EfficientNet model for DR classification. Kajan et al. (2020) [34] proposed a CNN model for identifying DR. They followed the TL technique for classifying the images. Patil et al. (2020) [35] employed a TL technique for DR-severity grading. Tariq et al. (2022) [36] employed ResNet50 and DenseNet121 models for the DR-severity-level classification model. They utilized APTOS and EYEPACS datasets for evaluating the model. Kobat et al. (2022) [37] applied a pre-trained DenseNet model to grade the DR-severity levels. Luo et al. (2023) [38] built a DR-detection model using deep CNN. They used local mining and long-range dependence techniques for the image classification. Lastly, Ishtiaq et al. (2023) [39] proposed a hybrid technique for classifying the fundus images.

To train deep-learning models, it is necessary to have access to extensive datasets that are both sizable and of superior quality. It can be challenging to obtain a diverse and representative dataset of retinal images, especially when dealing with rare DR conditions. The presence of imbalanced data may influence the model to produce more false positives [39]. It presents challenges in identifying severe instances of DR. Interoperability and user-friendly interfaces for healthcare professionals are essential to integrate DL models into clinical settings and EHR systems [39]. Processing retinal images in real time for prompt diagnosis and prioritization in telemedicine or point-of-care environments can impose a significant demand on resources and require specialist technology. The existing CNN models, including VGG, ResNet, and DenseNet models, demand huge computational resources for classifying DR severities [39]. There is a demand for lightweight applications to overcome the shortcomings of the existing models and to detect DR severities with limited computational resources.

## 3. Materials and Methods

The author presents a DL-based DR-severity grading model. MobileNet V3–Small is a lightweight CNN model that classifies complex images with fewer computational resources. However, the complexities in the fundus images may reduce the performance of the MobileNet V3. Integrating feature extraction and selection techniques enables the MobileNet V3 to produce optimal results and reduces the possibility of data overfitting. In addition, it minimizes the number of parameters of the model in learning the DR severity in the fundus images. The traditional feature-extraction and selection techniques demand a higher computational time for exploring the search space to reduce the dimensionality of the feature set. Yolo V7 [40] is the recent version of the Yolo techniques. It applies deep CNN in extracting the crucial features that represent DR severity. It processes the image at multiple layers and extracts hierarchical features. In addition, it can extract the key features in a short period. QMPA [41] is one of the optimization techniques that reduces the computation time in identifying the feature sets. Therefore, the author applies Yolov V7 and QMPA techniques in the proposed study for classifying DR severities using the fundus images.

Figure 1 highlights the proposed model for classifying the fundus images. Initially, the images were extracted from the APTOS [42] and EyePacs [43], which are benchmark datasets for DR. The author applies CLAHE and Wiener filter  functions to enhance the image quality. The Yolo V7 technique [42] is applied to extract the key features. QMPA [41] is modified to improve the performance in selecting the crucial features. In addition, an Adam Optimizer (AO) is used to optimize the hyperparameters of the MobileNet V3 model.

### 3.1. Data Acquisition

In this study, the author utilizes the APTOS and EyePacs datasets. The APTOS dataset is available in the repository [42]. It was generated by Aravind Eye Hospital, India. The clinicians captured the fundus images across India. The images were captured using multiple cameras. Thus, the images contain noise and artifacts. The EyePacs dataset is publicly available in the Kaggle repository [43]. It covers a larger number of fundus images. The images were collected from primary care centers across the USA. The dataset provider resized the images into 1024 × 1024 pixels and cropped the black spaces. Based on the severity, the clinicians rated each image as 0 (no DR), 1 (mild), 2 (moderate), 3 (severe), and 4 (proliferative DR). Table 1 presents the properties of the datasets and the definition of the notations used in this section is presented in Table 2. Figure 2 shows the sample images of the datasets.

### 3.2. Image Pre-Processing

To overcome the noise and artifacts, the author employs CLAHE and Wiener filter techniques. Firstly, CLAHE is used to improve the contrast and visibility of the fundus images. It divides the images into blocks and computes histograms for each block. A Wiener filter is applied to remove the noise from each pixel. It employs the frequency domain to reduce the mean square error between the original and reconstructed image. The transfer function is used to compute element-wise multiplication for removing the noise. Let I be the fundus image and WF be the Wiener filter function. Equation (1) computes the process of removing noise from the images.
(1)$$I=WF\;\left(I_{i}\right)\mathrm{where}i=1,\ldots ,N$$
where *i* = 1,..., N

Equation (2) presents the computation of error between the original and reconstructed image.
(2)$$e=k\left(X,Y\right)-\hat{k}\left(X,Y\right)$$

### 3.3. Data Augmentation

The author applied the rotation-range function to generate a set of images with a pre-defined range of degrees. Horizontal and vertical flips are used to produce randomly mirrored images. The author applies the shear-range method to distort the images to rectify the perception angles. In addition, width-shift and height-shift ranges are employed for shifting the images to horizontal and vertical positions, respectively. The proposed data-augmentation process is used to overcome the data imbalance of the dataset. The images are resized into 608 × 608 pixels, and each image is transformed into multiple angles. This process assists the training phase in providing an additional set of features to the CNN model.

### 3.4. Feature Extraction

The author employs Yolo V7 to extract the image features and generate the feature sets. It processes the textures of the fundus images in the lower layer and derives the semantic features in the higher layer. Figure 3 highlights the generation of the feature sets using the Yolo V7 technique.

In the feature map grid, Yolo V7 employs the detection head to compute the bounding boxes, likelihood of the object’s existence, and confidence score. Equation (3) shows the mathematical form of the feature set generation.
(3)$$F_{s}=Yolo\;V7\left(I_{i}\right)\mathrm{where}i=1,\ldots ,N$$

### 3.5. Modified Quantum Marine Predator Algorithm-Based Feature Selection

To select the key features from the feature sets, the author employs the QMPA algorithm. QMPA is a metaheuristic algorithm for selecting interesting features for DR-severity detection. It generates a feature set to support the following MobileNet V3 model. The feature set represents the presence of the useful feature. Equation (4) shows the initial feature set with size N.
(4)$$F_{i}\left(t+1\right)=F_{min}+r*\left(F_{max}-F_{min}\right)$$

QMPA derived Elite and Prey matrices from the traditional MPA to represent the predator with search strategy and the prey’s position data, respectively. However, QMPA faces challenges in achieving the global optima to obtain optimal features. Optimization algorithms frequently employ Cauchy and Gauss mutation [44] to improve efficiency. By introducing randomness into the population of potential solutions, these mutation operators broaden the search space and prevent the algorithm from being trapped in a local optimum. Thus, the author introduces the Cauchy–Gaussian mutation method to improve the searching strategy of the QMPA. Equations (5)–(7) outline the Cauchy–Gaussian mutation to achieve the global optimum.
(5)$$E\left(t+1\right)=E\left[1+\lambda_{1}Cauchy\left(0,\sigma^{2}\right)+\lambda_{2}Gauss\left(0,\sigma^{2}\right)\right]$$
(6)$$\sigma =exp\left(\frac{F\left(E\right)-F\left(E_{\alpha }\right)}{\left|F\left(E_{\alpha }\right)\right|}\right)$$
(7)$$F_{s}\left(t+1\right)=\left\{\begin{array}{c} E\left(t+1\right)f\left(E\left(t+1\right)\right)\leq f\left(E\right)\\ E,\;\;\;\;Otherwise \end{array}\right.$$

Furthermore, to find the best set of features, QMPA applies Equation (8) for computing the feature sets.
(8)$$F_{s}=\theta *KBest_{i}+\left(1-\theta \right)*MBest_{i}$$

QMPA computes the iteration using Equation (9).
(9)iteration→=RQ→⊗Elitei→−RQ→⊗Preyi→, i=1, …, N2

### 3.6. MobileNet V3–Small Model-Based DR-Severity Prediction

The author employs the MobileNet V3–Small model for classifying the fundus images. The MobileNet V3–Small neural-network architecture is the latest version of a series of networks developed for mobile and embedded devices. It has a versatile design enabling modification according to individual use cases with the trade-off between speed and accuracy. The MobileNet V3–Small architecture has been specifically designed and tuned to provide better inference performance and accommodate devices with limited computational resources. It incorporates the Hard swish activation function, a non-linear activation function that incorporates the favorable characteristics of the ReLu and sigmoid activation functions. The Hard swish function is specifically engineered to possess computing efficiency and exhibit a non-zero derivative at zero. It performs appropriate gradient-based optimization throughout the training process. Integrating squeeze and excitation (SE) blocks into MobileNet V3 is undertaken to augment channel-wise feature recalibration.

Utilizing SE blocks facilitates the adaptive scaling and recalibration of feature channels. It enables the network to prioritize the key features. The architectural design permits the incorporation of various configurations of layers and blocks under specific criteria. Equation (10) highlights the multi-class classification using the *MobileNet V3*–*Small* model.
(10)$$I_{C}=MobileNet\;V3–Small+ReLu\left(FC\left(FC\left(Softmax\left(F_{s}\right)\right)\right)\right)$$

Figure 4 shows the *MobileNet V3*–*Small* model for the DR-severity detection model.

Furthermore, AO is used to fine-tune the hyper-parameters of the Mobile-Net V3 model. Dropout layers are integrated with the classifier to achieve an optimal outcome based on the outcome.

### 3.7. Evaluation Metrics

The author employs the commonly used metrics, including accuracy, precision, recall, and F1-Score. Accuracy presents the model’s efficiency in correctly classifying the DR-severity levels. However, it may not be suitable for imbalanced datasets. Therefore, precision and recall are used to evaluate the model’s performance using true positives and true negatives. In addition, F1-Score provides a model’s performance based on false positives and false negatives. Equations (11)–(14) outline the computation of accuracy, precision, recall, and F1-Score.
(11)$$Accuracy=\frac{Number\;of\;correctly\;identified\;fundus\;images}{Total\;number\;of\;images}$$
(12)$$Precision=\frac{Number\;of\;correctly\;identified\;fundus\;images}{\left(Number\;of\;DR\;severity\;classes\right)+\left(Number\;of\;wrongly\;predicted\;fundus\;images\right)}$$
(13)$$Recall=\frac{Number\;of\;correctly\;identified\;fundus\;images}{\left(Number\;of\;DR\;severity\;classes\right)+\left(Number\;of\;wrongly\;predicted\;normal\;fundus\;images\right)}$$
(14)$$F1-Score=\frac{2\times Precision\times Recall}{\left(Precision+Recall\right)}$$

Cohen’s Kappa $\left(K\right)$ is used to find the relationship between the predicted and actual classifications. It measures the inter-rater reliability using true positives (*TP*), true negatives (*TN*), false positives (*FP*), and false negatives (*FN*). Equation (15) shows the mathematical expression for calculating K.
(15)$$K=\frac{2\times \left(TP\times TN-FN\times FP\right)}{\left(TP+FP\right)+\left(FP+TN\right)+\left(TP+FN\right)+\left(FN+TN\right)}$$

It is widely applied for measuring the efficiency of multi-class classification. Mean absolute deviation (MAD) and root mean square error (RMSE) are used to measure the model’s performance based on the actual observed values. The uncertainty levels are computed for the classifiers using confidence interval (CI) and standard deviation (SD). Equations (16) and (17) highlight the mathematical form of MAD and RMSE.
(16)$$\mathrm{MAD}=\frac{\sum \left|D-\mu \right|}{N}$$
(17)$$\mathrm{RMSE}=\sqrt{\frac{1}{N}\overset{N}{\underset{i=1}{\sum }}{\left(M-\hat{M}\right)}^{2}}$$

Furthermore, the author applies the model development settings to evaluate the computational complexities of the DR-severity models. The total number of parameters, learning rate, and floating point operations (FLOPs) are used to identify the model’s computational requirements for learning the key patterns of the fundus images. The testing time is used to find the model’s efficiency on the real-time images. Epochs (number of iterations) and the number of convolutional layers are used to evaluate the model’s capability to detect the DR severity. In addition, the ratio of input/output data is used to evaluate the model’s efficiency in handling the feature sets to predict the DR severity.

## 4. Results

In this study, the author implemented the proposed DR-detection model using Python 3.8.3, NVIDIA GeForce GTX, Windows 10 Professional, and Intel i7 processor with 3.2 GHz. The author generalized the proposed in APTOS and EyePacs datasets, respectively. The datasets are divided into training (70%) and testing (30%). Pytorch and Tensorflow libraries are employed for constructing the MobileNet V3 model. The MobileNet V3 model is optimized using Adam Optimizer (AO). The batch sizes of 54 and 86 and Epochs of 214 and 426 are used for APTOS and EyePacs datasets, accordingly. A softmax function, two dropouts, and three fully connected layers are added to the MobileNet V3 model. Table 3 highlights the performance of the proposed model in the APTOS dataset. The proposed model achieved a better outcome due to the feature extraction and selection techniques. In addition, the higher value of Kappa highlighted the significance of the proposed model in classifying multi-label images.

Likewise, Table 4 outlines the proposed model’s performance in the EyePacs dataset. Compared to the APTOS dataset, EyePacs covers a larger number of samples. The samples were used to train the proposed model to learn the crucial patterns of DR severity. The outcome highlighted that the proposed model obtained superior results. A higher value of F1-Score represents the model’s efficiency in dealing with true positives, true negatives, false positives, and false negatives. Figure 5 highlights the proposed model’s performance in APTOS and EyePacs datasets, respectively.

Table 5 highlights the findings of the comparative analysis. The proposed model achieved an exceptional outcome in the APTOS dataset. It achieved an average accuracy of 98.0 for the APTOS dataset. The APTOS dataset is highly imbalanced, and the proposed image pre-processing approach addressed the data imbalance by integrating high-quality images. In addition, Yolo V7 identified the tiny spots related to DR severity. Figure 6 presents the findings of the comparative analysis for the APTOS dataset.

Similarly, Table 6 reveals the performance of the DR-severity detection models in the EyePacs dataset. The EyePacs dataset presents the images at a high pixel rate. It favored the proposed model to resize the images without any compromise in the image quality. Yolo V7 identified the patterns, effectively. The Cauchy–Gaussian mutation has improved the computational efficiency of the suggested DR-severity detection model. The proposed model outperformed the existing models. Figure 7 shows the comparative analysis outcome for the EyePacs dataset.

Table 7 presents the computational strategies of the DR-severity detection models. The proposed model employed the MobileNet V3 model, which demands fewer parameters and FLOPs for image classification. Moreover, the MobileNet V3 model was trained using the ImageNet dataset. Thus, the proposed model obtained an optimal outcome in APTOS and EyePacs with fewer parameters and FLOPs. It reduces the computational complexities in classifying the DR severity using the fundus images. Thus, the proposed DR is a lightweight model that requires fewer parameters, a learning rate, and FLOPs to generate an exceptional outcome.

Table 8 outlines the findings of the loss-function analysis. It indicates that the proposed model obtained fewer errors in APTOS and EyePacs datasets. The feature extraction and selection approaches supported the proposed model to achieve an optimal outcome. The suggested model addressed the challenges in classifying the fundus images by integrating the Yolo V7 and QMPA models. Moreover, the inclusion of the Cauchy–Gaussian search strategy has played a significant role in the proposed model’s performance.

Finally, Table 9 highlights the uncertainty and variability of the proposed model’s efficiency in detecting DR severities. The proposed model achieved a better CI and SD for APTOS and EyePacs datasets, respectively. The higher value indicates the effective prediction in the unknown data. Moreover, the proposed model combined Yolo V7, QMPA, and MobileNet V3 models for image classification. The findings favored the integrated approach of the proposed model in detecting DR severity.

## 5. Discussions

In this study, the author proposed a DR-severity detection model for grading the severity of DR using fundus images. The proposed image processing process has produced high-quality images. Initially, the contrast was improved using the CLAHE technique. The author applied a Wiener filter to remove the noise. The data augmentation process has supported this proposed study to overcome the data imbalances in the APTOS dataset. In addition, it produced an additional number of training samples to train the proposed model. The fundus images undergo preprocessing techniques aimed at improving image quality and reducing noise interference. The model was trained with a special focus on the diagnosis of DR. It was trained using benchmark datasets. During the training process, the model acquired the ability to identify and extract pertinent characteristics from the fundus images. Based on the training, the proposed model detected the DR-severity levels from the real-time images.

In feature extraction, Yolo V7 provided the relevant features to the proposed classifier. It identified the crucial patterns of DR severity and generated the feature sets. The identified objects were collected as features. The author tailored the Yolo V7 model and retrieved the feature sets. The architecture of Yolo V7 has assisted the process of feature extraction to produce an outcome in a limited time. On the other hand, QMPA was used for the feature selection. The author introduced the Cauchy–Gaussian mutation searching strategy in the QMPA search space to improve the feature-selection process. Finally, the MobileNet V3–Small model classified the DR-severity levels using the feature sets. The author optimized the CNN model using an Adam optimizer. The model weights are iteratively modified to minimize the loss function that measures the discrepancy between the anticipated and the actual severity levels. Following the classification process, the proposed DR detects the degree of severity found in the fundus images.

Karki et al. [26] employed the EfficientNet model for DR-severity detection. They achieved a Kappa score of 92.4% in the EyePacs dataset. The EfficientNet model is the recently developed pre-trained image classifier. However, it requires an extended training time and a larger dataset for image classification. The complexity of the EfficientNet model may cause limitations in image classification. In contrast, the proposed model achieved a Kappa value of 91.1% with a lower computation cost.

Tariq et al. [29] proposed a DL technique for classifying the fundus images. They employed the ResNet 50 and DenseNet 121 models in the DR-severity classification. They obtained an accuracy of 63.0%. The CNN models face challenges in classifying images and demand additional training time. On the other hand, the proposed model is a lightweight application. It requires a small set of samples to learn the new environment.

Ishtiaq et al. [32] applied the local binary patterns for extracting the features. They employed the Binary Dragon Fly and Sine Cosine algorithms for optimizing the feature extraction process. They achieved an accuracy of 98.8 % in the EyePacs dataset. Similarly, the proposed model obtained an accuracy of 98.0%. However, the proposed model achieved a better Kappa value than the Ishtiaq et al. model.

Kobat et al. [30] proposed a DR-detection model using the pre-trained DenseNet model. They employed the horizontal and vertical patch division in extracting the features. They obtained an average accuracy of 84.9% and 86.7% in the APTOS and the EyePacs datasets. However, the proposed model outperformed the Kobat et al. model by achieving an exceptional outcome with fewer computational strategies.

Luo et al. [31] developed a DR-detection model using the deep CNN model. They employed long-range dependency among the lesion features for DR-severity detection. In addition, they followed patch-wise relationships to improve the local patch features. They obtained an average accuracy of 83.6% in the EyePacs dataset. In contrast, the proposed model detected the severity levels with higher accuracy.

The findings outlined that the proposed DR-severity detection model has the potential to play a role in the diagnosis and evaluation of the various severity levels associated with DR, a retinal disease that poses a risk to vision. It demonstrated a high level of suitability due to its exceptional proficiency in processing and evaluating the fundus images. The integration of CNN-based models into telemedicine and screening programs enables the streamlined and automated screening of extensive populations, hence enhancing efficiency. Immediate evaluation and therapy might be emphasized for patients who present more severe diabetic DR.

Healthcare and disease management greatly benefit from the proposed automated DR-severity detection system. These benefits aid in better patient outcomes, more effective healthcare delivery, and lower overall healthcare costs. The proposed model can identify DR in its earliest stages, typically when patients have any obvious symptoms. It can accurately and consistently analyze retinal images. To efficiently screen many patients, the suggested model can process a huge number of retinal pictures in a short amount of time. The application of the proposed model can substantially minimize the likelihood of human errors while interpreting retinal images. This practice improves the dependability of diagnoses and mitigates the likelihood of erroneous diagnostic assessments. The proposed model possesses the capability to speed up the delivery of outcomes and reduce the diagnostic duration. Telemedicine and other forms of remote healthcare delivery can utilize the suggested framework to provide DR screening to patients in underprivileged and remote areas. The proposed model can offer a reliable and uniform means of assessment, consequently reducing the potential for diagnostic discrepancies across diverse healthcare practitioners.

The proposed model produced a remarkable performance in DR identification and management. However, the author encountered limitations in classifying the fundus images using the proposed model. The expertise of ophthalmologists and other experts is still essential for deciphering AI-generated data, determining the best course of therapy, and caring for patients. The accuracy and dependability of the proposed model in clinical practice depend on thorough validation and ongoing improvement. A significant reference in the context of DR screening is microaneurysm. The dimensions of microaneurysms can be extremely small, rendering their detection challenging and susceptible to misidentification with other types of lesions. Additionally, the poor contrast between lesion pixels and background pixels, the irregular form of lesions, and the significant variations between the same lesion spots may cause limitations in diagnosing ophthalmic disorders. Thus, an effective image pre-processing technique is required for detecting DR severity in the real-time environment.

## 6. Conclusions

In this study, the author developed a multi-class DR-severity grading model using the DL technique. The proposed model integrated the image pre-processing, Yolo V7, QMPA, and MobileNet V3-Small models. The fundus image datasets are highly imbalanced. In addition, it contains noise and artifacts. The suggested image pre-processing technique has improved the image quality. The dataset biases were addressed using the data augmentation process. The feature extraction process applied the Yolo V7 technique to extract the key features. The author applied the QMPA with the Cauchy–Gaussian mutation strategy to select the critical features related to the DR severity. The MobileNet V3 model was employed to classify the images based on severity levels. The benchmark datasets, including APTOS and EyePacs, were used to generalize the proposed model. The findings highlight the significance of the proposed model in diagnosing DR severity. The proposed model offers an opportunity to develop a mobile-based application with which to treat DR patients. However, it encountered limitations in classifying the fundus images. The small dimension of DR severity in the fundus images may reduce the proposed model’s prediction accuracy. Effective image pre-processing is required to improve the quality of the real-time images. In the future, the author will extend the research to resolve the shortcomings of the proposed model.

## Figures and Tables

**Figure 1 diagnostics-13-03120-f001:**
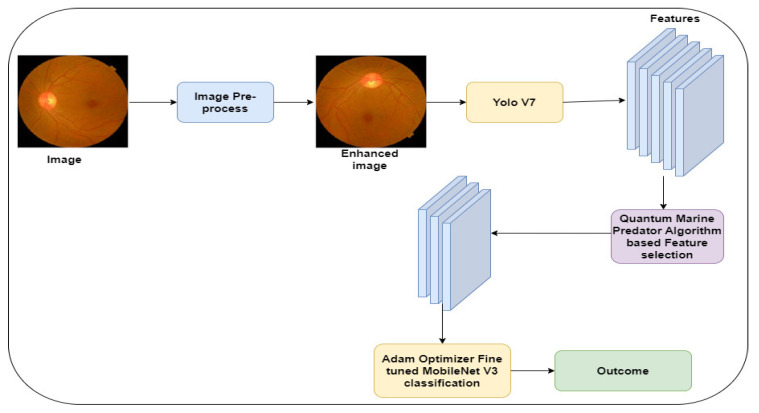
Proposed framework.

**Figure 2 diagnostics-13-03120-f002:**
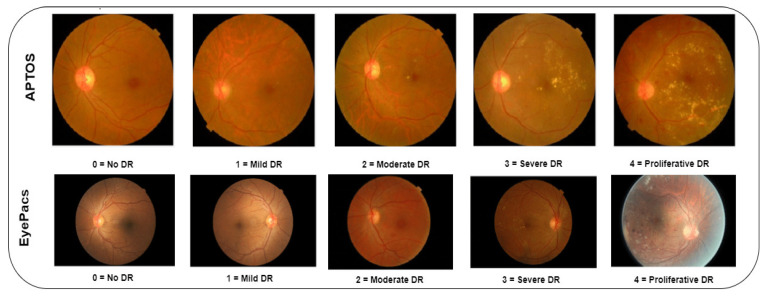
Sample images.

**Figure 3 diagnostics-13-03120-f003:**
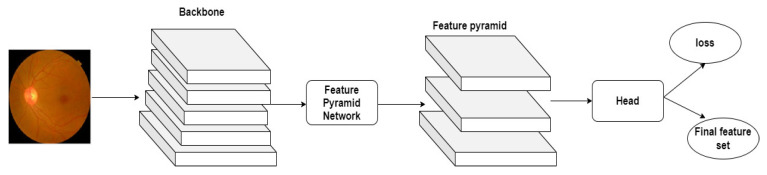
Feature-set generation.

**Figure 4 diagnostics-13-03120-f004:**
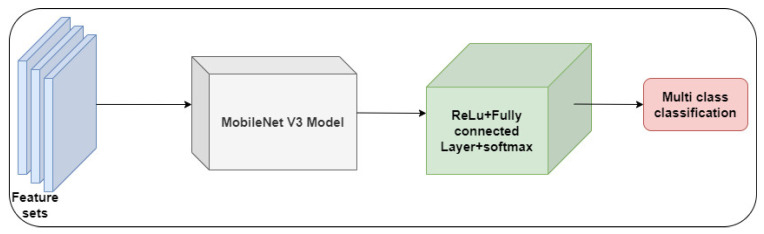
MobileNet V3—Multi class classification.

**Figure 5 diagnostics-13-03120-f005:**
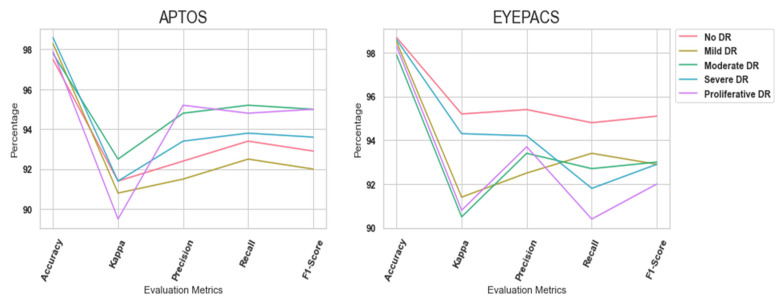
Performance analysis outcome.

**Figure 6 diagnostics-13-03120-f006:**
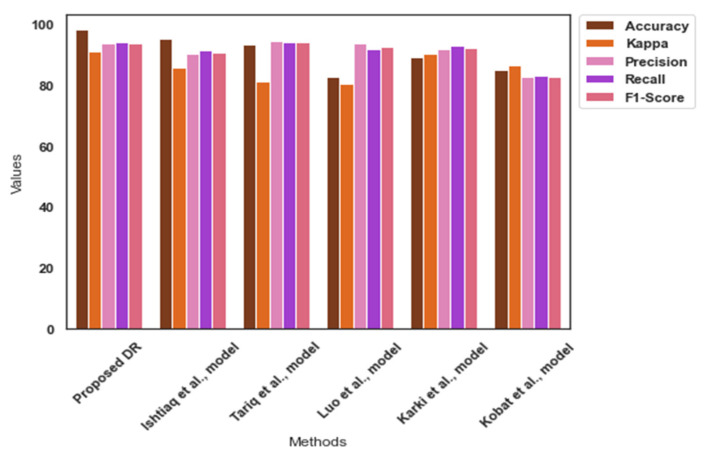
Comparative analysis findings—APTOS [33,36,37,38,39].

**Figure 7 diagnostics-13-03120-f007:**
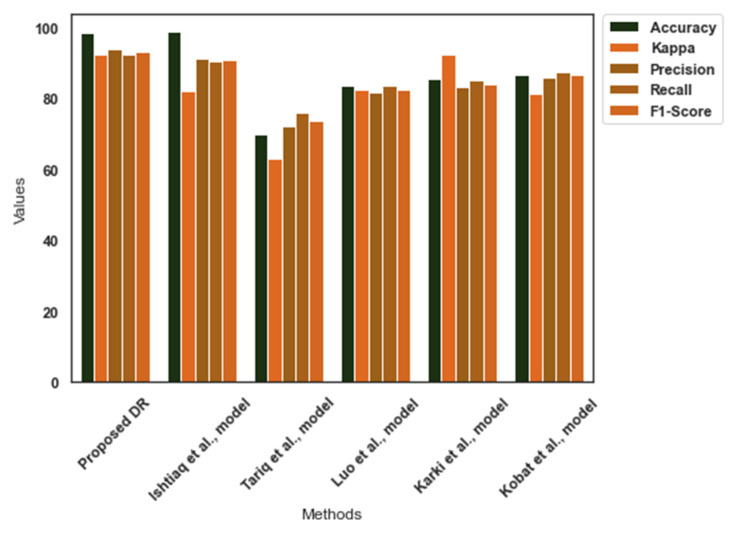
Comparative analysis findings—EyePacs [33,36,37,38,39].

**Table 1 diagnostics-13-03120-t001:** Dataset characteristics.

Dataset	Training	Testing
EyePacs	24,570	10,530
APTOS	3662	1928

**Table 2 diagnostics-13-03120-t002:** Notation and definition.

Notation	Definition
I	Fundus image
WF	Wiener filter
e	Mean square error
$k\left(X,Y\right)$	Original image with X and Y co-ordinates
$\hat{k}\left(X,Y\right)$	Reconstructed image with X and Y co-ordinates
$F_{s}$	Feature sets
Yolo_V7	Yolo V7 function
N	Number of images
$E\left(t+1\right)$	Post mutation position of Elite
E	Current position of Elite
$F\left(E\right)$	Fitness value of E
$F\left(E_{\alpha }\right)$	Fitness value of E at α
| |	The absolute value
$Cauchy\left(0,\sigma^{2}\right)$ and $Gauss\left(0,\sigma^{2}\right)$	Random variables of Cauchy and Gauss distribution with wavelet (σ)
θ	Quantum constant
KBest and MBest	Optimal feature sets in the specific iteration (*i*)
$\vec{R_{Q}}$	Chaotic number
$\vec{Elite_{i}}$ $\;\mathrm{and}\;\vec{Prey_{i}}$	Elite and Prey vectors in the specific iteration (*i*)
⊗	Element wise addition
$\lambda_{1}$ and $\lambda_{2}$	Dynamic parameters
$I_{C}$	Multi-class classification
ReLu	Rectified linear unit
Softmax	Softmax function for the multi-class classification
MobileNet V3−Small	MobileNet V3—Small model
FC	Fully connected layer
$\hat{M}$	Predicted class
M	Mean value of predicted class
D	Data point
μ	Mean

**Table 3 diagnostics-13-03120-t003:** Proposed DR performance analysis—APTOS.

Classes/Metrics	Accuracy	Kappa	Precision	Recall	F1-Score
0 (No DR)	97.5	91.4	92.4	93.4	92.9
1 (Mild DR)	98.3	90.8	91.5	92.5	92.0
2 (Moderate DR)	97.8	92.5	94.8	95.2	95.0
3 (Severe DR)	98.6	91.4	93.4	93.8	93.6
4 (Proliferative DR)	97.9	89.5	95.2	94.8	95.0
Average	98.0	91.1	93.4	93.9	93.7

**Table 4 diagnostics-13-03120-t004:** Proposed DR performance analysis—EyePacs.

Classes/Metrics	Accuracy	Kappa	Precision	Recall	F1-Score
0 (No DR)	98.7	95.2	95.4	94.8	95.1
1 (Mild DR)	98.5	91.4	92.5	93.4	92.9
2 (Moderate DR)	97.9	90.5	93.4	92.7	93.0
3 (Severe DR)	98.6	94.3	94.2	91.8	92.9
4 (Proliferative DR)	98.3	90.8	93.7	90.4	92.0
Average	98.4	92.4	93.8	92.6	93.1

**Table 5 diagnostics-13-03120-t005:** Findings of comparative analysis—APTOS.

Methods/Metrics	Accuracy	Kappa	Precision	Recall	F1-Score
Proposed DR	98.0	91.1	93.4	93.9	93.7
Ishtiaq et al. model [39]	95.2	85.6	90.1	91.4	90.7
Tariq et al. model [36]	93.0	81.2	94.5	93.8	94.1
Luo et al. model [38]	82.4	80.4	93.4	91.8	92.5
Karki et al. model [33]	89.1	90.1	91.6	92.7	92.1
Kobat et al. model [37]	84.9	86.4	82.4	83.1	82.7

**Table 6 diagnostics-13-03120-t006:** Findings of comparative analysis—EyePacs.

Methods/Metrics	Accuracy	Kappa	Precision	Recall	F1-Score
Proposed DR	98.4	92.4	93.8	92.6	93.1
Ishtiaq et al. model [39]	98.8	82.3	91.2	90.7	90.9
Tariq et al. model [36]	70.0	63.0	72.0	76.0	73.9
Luo et al. model [38]	83.6	82.4	81.9	83.5	82.6
Karki et al. model [33]	85.4	92.4	83.4	85.2	84.2
Kobat et al. model [37]	86.7	81.4	86.1	87.3	86.7

**Table 7 diagnostics-13-03120-t007:** Computational strategies.

Methods	APTOS 2019			EyePacs		
	Learning Rate	Parameters (in Millions (m))	FLOPs (in Giga (G))	Learning Rate	Parameters (in Millions (m))	FLOPs (in Giga (G))
Proposed DR	1 × 10^−4^	47 M	2.3 G	1 × 10^−3^	77 M	4.5 G
Ishtiaq et al. model [39]	1 × 10^−3^	86 M	4.7 G	1 × 10^−2^	94 M	5.1 G
Tariq et al. model [36]	1 × 10^−3^	72 M	4.3 G	1 × 10^−3^	98 M	5.6 G
Luo et al. model [38]	1 × 10^−3^	64 M	3.7 G	1 × 10^−2^	97 M	5.9 G
Karki et al. model [33]	1 × 10^−3^	57 M	4.1 G	1 × 10^−2^	91 M	5.3 G
Kobat et al. model [37]	1 × 10^−2^	71 M	3.9 G	1 × 10^−2^	89 M	4.9 G

**Table 8 diagnostics-13-03120-t008:** Outcome of loss-function analysis.

Methods	APTOS			EyePacs		
	MAD	RMSE	Testing Time (seconds)	MAD	RMSE	Testing Time (seconds)
Proposed DR	0.385	0.754	1.26	0.423	0.821	1.38
Ishtiaq et al. model [39]	0.425	0.823	1.83	0.518	0.914	1.45
Tariq et al. model [36]	0.398	0.912	2.31	0.467	0.965	1.52
Luo et al. model [38]	0.405	0.864	2.42	0.523	0.974	1.69
Karki et al. model [33]	0.512	0.845	2.51	0.612	1.012	1.54
Kobat et al. model [37]	0.487	1.021	2.35	0.724	1.125	2.24

**Table 9 diagnostics-13-03120-t009:** Uncertainty analysis.

Methods	APTOS		EyePacs	
	CI	SD	CI	SD
Proposed DR	[97.53–97.61]	0.0014	[98.32–98.56]	0.0019
Ishtiaq et al. model [39]	[95.80–96.32]	0.0022	[95.68–96.18]	0.0028
Tariq et al. model [36]	[96.30–97.12]	0.0021	[95.24–95.86]	0.0032
Luo et al. model [38]	[95.83–95.89]	0.0016	[96.40–97.12]	0.0017
Karki et al. model [33]	[97.10–97.45]	0.0017	[97.19–98.40]	0.0018
Kobat et al. model [37]	[97.42–97.65]	0.0019	[96.58–96.68]	0.0021

## Data Availability

Dataset. Available online: https://www.kaggle.com/c/aptos2019-blindness-detection (accessed on 23 May 2023). Foundation Consumer Healthcare. EyePACS: Diabetic Retinopathy Detection. Available online: https://www.kaggle.com/c/diabetic-retinopathy-detection/data (accessed on 25 May 2023).

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
