# Peer review of "A Lightweight Diabetic Retinopathy Detection Model Using a Deep-Learning Technique"

_diagnostics, 2023, doi:10.3390/diagnostics13193120_

Round 1
Reviewer 1 Report
The manuscript shows a complete, clear and well organized presentation. The background section of the paper demonstrates a clear relationship to the problem. Enough literature has been provided. The sources cited are original, authoritative, important, and recent. The following reviews would make study more exciting and clear.
1- The resolution of Figure 5, Figure 6 and Figure 7 should be improved.
2- Equations of evaluation metrics should be given.
3- MobileNet V3 has been written in different ways (MobileNet V3, MobileNetV3). They should all be written in the same format.
The manuscript shows a complete, clear and well organized presentation. The background section of the paper demonstrates a clear relationship to the problem. Enough literature has been provided. The sources cited are original, authoritative, important, and recent. The following reviews would make study more exciting and clear.
1- The resolution of Figure 5, Figure 6 and Figure 7 should be improved.
2- Equations of evaluation metrics should be given.
3- MobileNet V3 has been written in different ways (MobileNet V3, MobileNetV3). They should all be written in the same format.
Author Response
Dear Editor and Reviewers,
I thank Editor and Reviewers for their suggestions in improving the standard of the article. I addressed the reviewers’ comments and modified the article.
Reviewer 1:
The manuscript shows a complete, clear and well organized presentation. The background section of the paper demonstrates a clear relationship to the problem. Enough literature has been provided. The sources cited are original, authoritative, important, and recent. The following reviews would make study more exciting and clear.
1- The resolution of Figure 5, Figure 6 and Figure 7 should be improved.
Response: As per the suggestion, I changed the resolutions of the figures to 600 dpi.
2- Equations of evaluation metrics should be given.
Response: As per the suggestion, I introduced the evaluation metrics in line no. 332.
3- MobileNet V3 has been written in different ways (MobileNet V3, MobileNetV3). They should all be written in the same format.
Response: As per the suggestion, I updated the article with same term for MobileNet V3.

Reviewer 2 Report
Manuscript Title: A lightweight Diabetic Retinopathy Detection Model using Deep Learning Technique
Overview: In this paper the authors proposed a novel model for DR (Diabetic Retinopathy) diagnosis using deep learning on fundus images. Overall, the article is well written. The problem is unique and important. The methods and material need some justifications (see the comments below). The results are briefly explained. Discussions have valid arguments, and the conclusion is convincing. The overall merit of the paper is good and can be accepted after a single revision round, nevertheless I have some suggestions to improve its quality.
Comments:
· The introduction section contains the existing work on DR done by the research community, that is good. However, it may be enhanced by discussing recent trends in medical image analysis related to ophthalmology in general and then focus on retinopathy detection like diabetic and hypertensive retinopathy.
· Yet, I understand that there are few works on this area, but it is suggested to separate the literature review (technical) from the introduction section by exploring and adding more (technical) works on retinopathy.
· The list of contributions is not appealing to be enough for an original article. Pls revise and enhance them, same should be reflected in abstract. Another suggestion is that the contributions can be shown in other ways, like comparison of activation functions, their selection criterion or sensitivity analysis, etc.
· The figures’ quality can be enhanced using a high resolution of at least 600 dpi.
· Some figures are of very low quality and are blur, e.g fig 5,6 and 7. Pls redraw them.
· The lightweight model in the title portrays that the computational complexity (spatial or temporal) will be minimized. I can’t` see anything on complexity reduction technique in methods and materials, and complexity analysis with other models in the results.
· What is the motivation to use Yolo V7. I believe for fundus images of retinopathy used in your study, the CNNs are well suited due to their inherited properties (Spatial Hierarchies, Translation Invariance, Pretrained Models) e.g., VGG, ResNet, DenseNet etc, being the standard choices for medical image analysis. Why are they not considered, there must be some reason? This justification is required in the manuscript and author response file.
· The authors said they use modified QMPA meta heuristics technique for feature selection. Aren’t there any other techniques that can do this? If yes, then why they are not used? Moreover, there are deep models that have built in capability of feature selection as I mentioned above. Justify the usage of QMPA.
· Similarly for feature classification MobileNet V3 is used during model training aided with ReLU and Softmax activation functions, despite the models I listed above can do this by utilizing same activation functions or may be some other like Sigmoid or Regresion.
· I feel the authors have developed their own framework in which they adapt (Yolo V7+QMPA+MobileNet V3). A major motivation for this is required, as commented above.
· Add a table of symbols and notations for equations used in the paper.
· The article needs minor English proof reading. The list of references is sufficient.
· In section 2.7 the authors discussed evaluation metrics such as RMSE and MAD, which is good. In addition, add another subsection about the details on model development settings, tuning parameters, learning rate, number of iterations, sample size, ratio of input-to-output data during learning, testing data, training time, testing time, machine specifications, etc.
The English language seems fine, but a single round of proof reading will be appreciated.
Author Response
Dear Editor and Reviewers,
I thank Editor and Reviewers for their suggestions in improving the standard of the article. I addressed the reviewers’ comments and modified the article.
The introduction section contains the existing work on DR done by the research community, that is good. However, it may be enhanced by discussing recent trends in medical image analysis related to ophthalmology in general and then focus on retinopathy detection like diabetic and hypertensive retinopathy.
Response: As per the suggestion, I have included a paragraph in line no. 49. To present the recent techniques for treating DR.
- Yet, I understand that there are few works on this area, but it is suggested to separate the literature review (technical) from the introduction section by exploring and adding more (technical) works on retinopathy.
Response: As per the suggestion, I introduced the literate review in line no.126.
- The list of contributions is not appealing to be enough for an original article. Pls revise and enhance them, same should be reflected in abstract. Another suggestion is that the contributions can be shown in other ways, like comparison of activation functions, their selection criterion or sensitivity analysis, etc.
Response: As per the suggestion, I changed the contributions in the introduction part and modified the abstract.
- The figures’ quality can be enhanced using a high resolution of at least 600 dpi.
- Some figures are of very low quality and are blur, e.g fig 5,6 and 7. Pls redraw them.
Response: Thank you for the suggestion. I improved the quality of the figures.
- The lightweight model in the title portrays that the computational complexity (spatial or temporal) will be minimized. I can’t` see anything on complexity reduction technique in methods and materials, and complexity analysis with other models in the results.
Response: Thank you for the suggestion. I introduced the feature extraction and selection techniques to reduce the dimensionality of the features. This process will support the CNN model to overcome image complexities and produce results with limited computational resources. Table 7 highlights the computational complexities of the DR severity detection models. The proposed model required less number of parameters, learning rate, and computation time for generating the outcome.
- What is the motivation to use Yolo V7. I believe for fundus images of retinopathy used in your study, the CNNs are well suited due to their inherited properties (Spatial Hierarchies, Translation Invariance, Pretrained Models) e.g., VGG, ResNet, DenseNet etc, being the standard choices for medical image analysis. Why are they not considered, there must be some reason? This justification is required in the manuscript and author response file.
Response: I introduced a motivation for selecting Yolo v7, QMPA and MobileNet V3 in line no. 198. Yolo V7 identifies the key features with high accuracy. QMPA reduces the computation time of the MobileNet V3 model by reducing the dimensionality of the features.
- The authors said they use modified QMPA meta heuristics technique for feature selection. Aren’t there any other techniques that can do this? If yes, then why they are not used? Moreover, there are deep models that have built in capability of feature selection as I mentioned above. Justify the usage of QMPA.
Response: QMPA is quantum-based feature selection technique. It reduces the dimensionality of the features and minimizes the computation time.
- Similarly for feature classification MobileNet V3 is used during model training aided with ReLU and Softmax activation functions, despite the models I listed above can do this by utilizing same activation functions or may be some other like Sigmoid or Regresion.
Response: I thank you for your comments. MobileNet V3 is a lightweight CNN model. However, less number of convolutional layer may reduce the classification accuracy while dealing with larger number of features. Thus, I employed Yolo V7 and QMPA for improving the model’s accuracy.
- I feel the authors have developed their own framework in which they adapt (Yolo V7+QMPA+MobileNet V3). A major motivation for this is required, as commented above.
Response: I introduced a motivation for selecting Yolo v7, QMPA and MobileNet V3 in line no. 198.
- Add a table of symbols and notations for equations used in the paper.
Response: I introduced table 2 for presenting the notation with description in line no. 235.
- The article needs minor English proof reading. The list of references is sufficient.
Response: Thank you for your suggestion. As per the suggestion, the proofread was carried out by Dr. Arunadevi, Professor, Cambridge Institute of technology, Bengaluru, India.
- In section 2.7 the authors discussed evaluation metrics such as RMSE and MAD, which is good. In addition, add another subsection about the details on model development settings, tuning parameters, learning rate, number of iterations, sample size, ratio of input-to-output data during learning, testing data, training time, testing time, machine specifications, etc.
Response: As per the suggestion, I introduced a sub-section in line no. 353 for presenting the model development settings.

Reviewer 3 Report
Globally, the manuscript is very well written and organized.
However, there are some corrections and issues that should be addressed.
First, the English needs some corrections; please refer to the attached commented PDF document where some of the needed corrections are highlighted. I also recommend the author to carefully re-ready the full manuscript.
Define every acronym/abbreviation the first time you use it.
Use math style whenever you are referring to mathematical symbols/quantities.
Fully and carefully explain the data augmentation techniques you have used/employed.
Clearly list and explain the limitations of the study.
Lastly, please refer to the attached commented PDF document to address some other minor issues.

Please refer to the comments above.
Author Response
Dear Editor and Reviewers,
I thank Editor and Reviewers for their suggestions in improving the standard of the article. I addressed the reviewers’ comments and modified the article.
Globally, the manuscript is very well written and organized.
However, there are some corrections and issues that should be addressed.
First, the English needs some corrections; please refer to the attached commented PDF document where some of the needed corrections are highlighted. I also recommend the author to carefully re-ready the full manuscript.
Response: I thank you for your suggestion. As per the suggestion, I corrected the article as per the attached PDF file.
Define every acronym/abbreviation the first time you use it.
Response: I introduced the acronym for the first time as per the suggestion.
Use math style whenever you are referring to mathematical symbols/quantities.
Response: I changed the letters using the math style.
Fully and carefully explain the data augmentation techniques you have used/employed.
Response: As per the suggestion, I included a paragraph in line no. 249.
Clearly list and explain the limitations of the study.
Response: As per the suggestion, the limitations of the proposed study is presented in line no. 507
Lastly, please refer to the attached commented PDF document to address some other minor issues.
Response: Thank you for your suggestion. I addressed the minor issues as listed in the PDF.
